# An Improved Eclat Algorithm Based on Tissue-Like P System with Active Membranes

**Linlin Jia, Laisheng Xiang and Xiyu Liu \***

Business School, Shandong Normal University, Jinan 250358, China

**\*** Correspondence: xyliu@sdnu.edu.cn

**Abstract:** The Eclat algorithm is a typical frequent pattern mining algorithm using vertical data. This study proposes an improved Eclat algorithm called ETPAM, based on the tissue-like P system with active membranes. The active membranes are used to run evolution rules, i.e., object rewriting rules, in parallel. Moreover, ETPAM utilizes subsume indices and an early pruning strategy to reduce the number of frequent pattern candidates and subsumes. The time complexity of ETPAM is decreased from $O(t^2)$ to $O(t)$ as compared with the original Eclat algorithm through the parallelism of the P system. The experimental results using two databases indicate that ETPAM performs very well in mining frequent patterns, and the experimental results using four databases prove that ETPAM is computationally very efficient as compared with three other existing frequent pattern mining algorithms.

**Keywords:** frequent pattern mining; eclat algorithm; tissue-like P systems; membrane computing

---

## 1. Introduction

Membrane computing is a branch of natural computing, and its development provides many computing frameworks and new bio-molecular computing models [1]. The development of membrane computing started with observing the structure and functions of living cells. Membranes play important roles in the functioning of cells and separate the cells from the outside environment [2,3]. The models extracted from membrane computing are usually called P systems and are divided into three main categories, i.e., cell-like P systems, neural-like P systems, and tissue-like P systems. Most of these P systems have high computing power, are very efficient and Turing universal [4]. This study employs a tissue-like P system with active membranes to mine frequent patterns. A P system mainly consists of three parts: Membrane structure, multiple sets of objects, and evolution rules. For the membrane structure, the size and spatial layout are not important, and the focus is on the relationship between membranes [5,6]. Object sets are usually represented by a string of symbols and evolution rules of the objects are given in the form of rewriting rules. A P system is a distributed and parallel computing model, and evolutionary rules run synchronously, non-deterministically, and in maximum parallel [7], making the system computationally very efficient.

Data mining is a knowledge discovery process from large amounts of data and has been extensively studied in many fields. Frequent pattern mining is a fundamental field of data mining, and the goal is to find patterns that appear frequently in a database [8–11]. Many algorithms for mining frequent patterns, such as Apriori, FP-growth, and Eclat, to mention only a few, have been developed. Apriori utilizes an iterative approach called level-wise search, where $k + 1$ itemsets are generated from $k$ itemsets, by taking the join and prune actions [12,13]. Nevertheless, the database must be scanned multiple times, which is inefficient for large-scale databases. FP-growth employs an FP-tree structure and pattern fragment growth method to mine frequent patterns [14,15], but it is difficult to generate a main memory-based FP-tree when the database is large. Eclat mines frequent patterns using vertical data

different from Apriori and FP-growth, and only needs to read the columns relevant to the query to avoid reading unnecessary columns. In the Eclat algorithm, a new candidate set is generated from the union of two sets. By finding the intersection of the *TID_sets* of the two itemsets, the support count of the candidate set is quickly obtained. However, when there are too many candidates, the following problems will occur: (i) The operation of finding the intersection of the *TID_sets* is time consuming; and (ii) the scale of the *TID_set* is quite large and consumes a lot of memory. Many important improvements have been proposed [16–18]. However, it is necessary to improve the computational efficiency of the Eclat algorithm when the database becomes large.

This study proposes an improved Eclat algorithm called ETPAM based on the tissue-like P system with active membranes. The active membranes are used to generate subsume indices and frequent patterns. ETPAM utilizes the parallelism of the P system to execute rules in parallel. For a database with $t$ items, the algorithm generates $t + 1$ cells, uses $t$ cells to explore frequent patterns, and uses the other cell, usually cell 0, as the output cell to output all frequent patterns generated. The subsume is considered as a technique that can greatly reduce the size of the search space [19], the priori law is introduced into the frequent mining process, and a threshold is used to limit the number of candidates of subsumes to further improve efficiency. The time complexity of ETPAM is reduced from $O(t^2)$ to $O(t)$ as compared with the original Eclat algorithm. Experimental results using two databases indicate that ETPAM performs very well in frequent pattern mining, and those using four databases shows that ETPAM is computationally very efficient as compared with three existing frequent pattern mining algorithms.

The rest of this paper is arranged as follows. Section 2 describes the frequent pattern mining problem, the original Eclat algorithm, and the basic tissue-like P system. Section 3 introduces the design of the tissue-like P system for ETPAM, and provides explanations of the rules and the computing process. Section 4 presents an example to show how ETPAM works. In Section 5, two databases are used to evaluate the performance of the tissue-like P system in identifying frequent patterns and four databases are used to verify the efficiency of ETPM. Conclusions are drawn and further research directions are given in Section 6.

## 2. Preliminaries

In this section, some basic definitions about frequent pattern mining [10,11], the original Eclat algorithm, and structure of the tissue-like P system with active membranes are introduced.

### 2.1. Frequent Pattern Mining

Let $I = \{I_1, I_2 \ldots I_n\}$ be a set of items and $DB = \{T_1, T_2 \ldots T_m\}$ be a transaction database with $m$ transactions.

(i)   *Pattern: A set of items $P \subseteq I$ is called a pattern or an itemset.*
(ii)  *h-pattern: A pattern consisting of $h$ items.*
(iii) *Support count: The number of transactions containing a certain pattern P, denoted as $sup(P)$.*
(iv)  *Frequent pattern: A pattern with a support count no less than a given threshold k is called a frequent pattern.*

### 2.2. The Eclat Algorithm

Eclat mines frequent patterns using the vertical data format [18,20] that is different from Apriori and FP-growth because they use horizontal data.

Vertical data: The more commonly used horizontal data is in a format *TID*: Itemset, where *TID* represents a unique transaction $T_1, T_2, \ldots, T_m$ in a transaction database *DB*, and an itemset represents a set of items $I_1, I_2, \ldots, I_n$ that belong to a transaction. Relatively, vertical data is in a format of item: *TID_set*, where item represents the unique item $I_1, I_2 \ldots I_n$ in itemset *I*, and *TID_set* represents the set of transactions $T_1, T_2 \ldots T_m$ that include the corresponding item. An example of vertical database is

shown in Table 1. Vertical data is more efficient than horizontal data in the process of obtaining the support of items because an algorithm only needs to read the columns related to a query, but does not need to read other unnecessary columns. For instance, if the support of itemset $\{I_1 \ I_2\}$ is needed in Table 1, an algorithm just needs to read and intersect the *TID_sets* of $I_1$ and $I_2$ and find support $(I_1 \ I_2)$ = Num[(1, 4, 5, 7, 8, 9)∩(1, 2, 3, 4, 6, 8, 9)] = Num(1, 4, 8, 9) = 4, instead of scanning the entire database as using horizontal data.

**Table 1.** A transaction database.

| Item | TID_Set |
|------|---------|
| $I_1$ | $T_1 \ T_4 \ T_5 \ T_7 \ T_8 \ T_9$ |
| $I_2$ | $T_1 \ T_2 \ T_3 \ T_4 \ T_6 \ T_8 \ T_9$ |
| $I_3$ | $T_3 \ T_5 \ T_6 \ T_7 \ T_8 \ T_9$ |
| $I_4$ | $T_1 \ T_2 \ T_4$ |
| $I_5$ | $T_1 \ T_8$ |

The basic Eclat algorithm is described as follows and the procedure using the example database in Table 1 is shown in Figure 1.

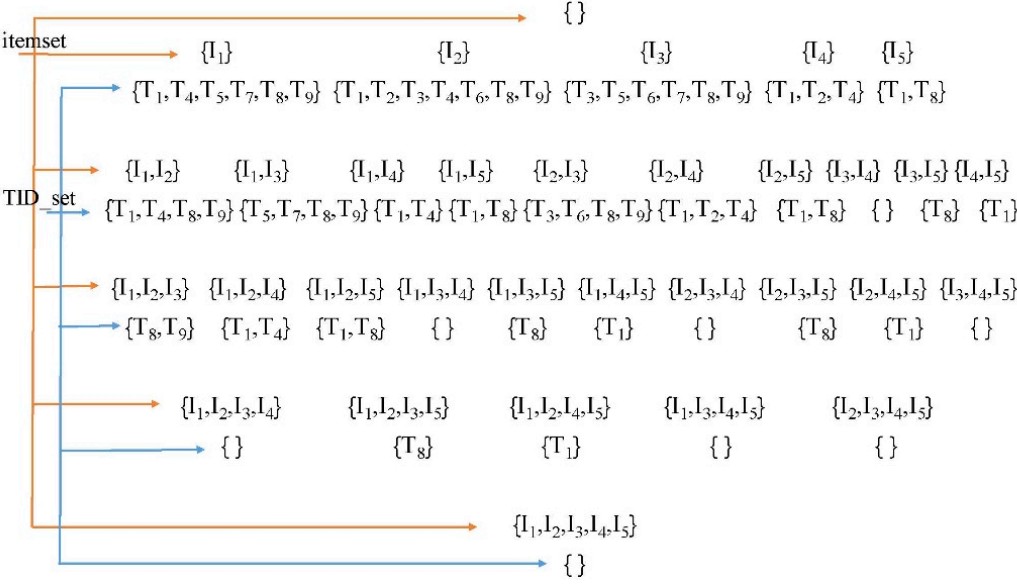

**Figure 1.** The procedure of Eclat for the example database in Table 1.

*Input*: Database in vertical data format and the threshold *k*.

*Step 1*: Take all items as a set $I_{all}$ and find all subsets of the set $I_{all}$. Let subset *i* be $I_i$, as shown in the labeling process in the red arrow in Figure 1.

*Step 2*: Find the intersection of each pair of the transaction sets *TID_sets* corresponding to the items in each subset, let intersection *i* be denoted by $T_i$, as shown in the labeling process in the blue arrow in Figure 1.

*Step 3*: Count the number of items in each $T_i$, and find the support of each itemset. Itemsets with support count greater than or equal to the threshold *k* are frequent itemsets.

*Output*: Frequent patterns with support not less than the threshold *k*.

### 2.3. Tissue-Like P Systems

The tissue-like P system is an important expansion of the cell-like P system [21]. In a tissue-like P system, multiple cells are placed in the same environment, both cells and the environment can contain objects, and the cells and the environment communicate through evolution rules. Evolution

rules are conducted in a non-deterministic and maximally parallel manner, and usually can produce an exponential growth space within linear operation steps [22]. When no evolution rules can be executed, the operation of the system stops and the final results are stored in a specific cell.

A basic tissue-like P system is a construct of the form:

$$\Pi = \{O, \sigma_1, \sigma_2, \ldots, \sigma_m, \text{syn}, \rho, i_{out}\},$$

where:

(i)     $O$ is a non-empty alphabet that represents a collection of objects in the tissue-like P system.

(ii)    syn $\subseteq \{1, 2, \ldots, m\} * \{1, 2, \ldots, m\}$ represents all channels between cells.

(iii)   $\rho$ represents the execution order of the rules in the membranes.

(iv)    $i_{out}$ is the output membrane which stores the final results of the algorithm.

(v)     $\sigma_1, \sigma_2, \ldots, \sigma_m$ are the cells, each of which is a construct of the form:

$$\sigma_h = (w_{h,0}, R_h), 1 \leq h \leq m,$$

where $w_{h,0}$ is the object set initially in cell $h$, if no object is in cell $h$ initially, $w_{h,0}$ is empty represented by $\lambda$, and $R_h$ is the set of evolution rules in cell $h$. A rule $R_h$: $u_\epsilon \rightarrow vw_{go}$ means removing the object multiset represented by $u$, generating the object multiset represented by $v$ and $w$, and sending the objects in $v$ and $w$ out to a specific area according to the target command. In the rule, $w_{go}$ means objects in $w$ are sent to the cells connected to the current cell, and v means objects in $v$ stay in the current cell. In $u_\epsilon$, $\epsilon$ is the promoter of the rule. If $u_{\neg\epsilon}$ is in the rule, $\neg\epsilon$ is the inhibitor of the rule. If the rule has a promoter, the rule can be executed only when all objects in the promoter appear, and if the rule has an inhibitor, the rule cannot be executed when the objects in the inhibitor appear. Active membranes are used to generate subsume indices for frequent 1-patterns, and dissolved when all subsume indices are found.

## 3. The ETPAM Algorithm

This section begins with an introduction of two improvements to the Eclat algorithm. The design of the tissue-like P system with active membranes to improve the algorithm is then discussed. The evolution rules and the computing process are explained next.

### 3.1. Improvements to the Eclat Algorithm

Improvement 1: The subsume index and a quick method to generate it. The subsume index is used to restrict the number of candidates in the process of frequent pattern mining [19,23].

Definitions: *subsume* $(A)$ represents the subsume index of pattern $A$:

$$subsume\,(A) = \{B \in I \mid g\,(A) \subseteq g\,(B)\}.$$

$g\,(A)$ represents the set of transactions $T_1, T_2 \ldots T_m$ including pattern $A$.

Property: $sup\,(A) = sup(A \cup S)$, $\forall\ S \in \{$subsets of $subsume\,(A)\}$.

The support of pattern $A$ is the same as the support of the union of the patterns that are subsets of $subsume\,(A)$ with pattern $A$.

Eclat mines frequent patterns using the vertical data format. $g\,(A)$ is the *TID_set* of transactions, including pattern $A$. Using vertical data can generate subsumes of 1-patterns quickly and effectively.

Improvement 2: Early pruning of the search space by the threshold. In step 1 of the Eclat algorithm, all items are taken as the set $I_{all}$ and all subsets of set $I_{all}$ are found. This step generates too many candidate subsets when the size of *TID_set* is large. Hence, a priori law is introduced to prune the search space early. In the process of obtaining the $(h + 1)$-itemsets through the intersections of the frequent $h$-itemsets, a $h$-itemset with a support count not larger than the threshold will be removed

from the intersection since any superset containing this itemset cannot be a frequent itemset and candidates containing this itemset do no need to be generated. The process of generating subsume indices is also improved. Just finding subsume indices for items with support counts larger than the threshold instead of subsume indices for all items in the database reduces the time and memory used.

## 3.2. Algorithm and Evolution Rules

Assume that the database *DB* contains $N$ transactions with $t$ fields. The tissue-like P system with active membranes with $t + 1$ cells, designed for the ETPAM algorithm, is shown in Figure 2. Frequent patterns are generated in cells 1 to $t$. The union of each of these frequent patterns and its corresponding subsume indices are formed in cell 0, where all frequent patterns are finally obtained.

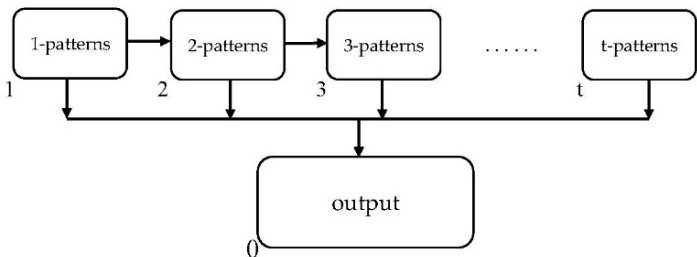

**Figure 2.** The tissue-like P system for the ETPAM algorithm.

The threshold $k$ is represented by $\theta^k$ in the P system. An object $T^i_j$ represents a transaction $T_i$ containing item $I_j$. In this way, the vertical database can be transformed into objects used in the P system. Auxiliary object $\beta$ is used to perform the comparison between the support of an itemset and the threshold. In the comparison, one item in the itemset consumes one $\beta$, and $\beta^k_j$ means $k$ copies of object $\beta_j$. Object $\xi$ is the promoter of the intersection process, and the corresponding rules can be executed only when object $\xi$ appears. Object $\zeta$ is the catalyst to delete the redundant $T^i_{j_1 j_2 \ldots j_t}$ in cell $t$, keeping the uniqueness of the object.

The tissue-like P system with active membranes for ETPAM is defined as follows:

$$\Pi = \{O,\ \sigma_1,\ \sigma_2, \ldots, \sigma_{t+1},\ syn,\ \rho,\ i_{out}\},$$

where:

(i)    $O = \{T^i_j,\ T^i_{j_1 j_2}, \ldots, T^i_{j_1 \ldots j_t},\ \beta_j,\ \beta_{j_1 j_2}, \ldots \beta_{j_1 \ldots j_t},\ X^{j_1}_{j_2},\ \xi,\ \zeta\}$, for $1 \leq i \leq N$ $1 \leq j_1 \leq j_2 \ldots j_t \leq t$;

(ii)   $syn = \{\{0,1\},\ \{0,2\}, \ldots, \{0,t\};\ \{1,2\},\ \{2,3\}, \ldots, \{t-1,t\}\}$;

(iii)   $\rho = \{r_i > r_j | i < j\}$;

(iv)   $\sigma_0 = (w_{0,0},\ R_0),\ \sigma_1 = (w_{1,0},\ R_1), \ldots, \sigma_t = (w_{t,0},\ R_t)$;

(v)   $i_{out} = 0$.

In $\sigma_0 = (w_{0,0},\ R_0)$, $w_{0,0} = \{\lambda\}$ and

$$R_0 = \{I_{j_1} I_{j_1 j_2}, \ldots, I_{j_1 j_2 \ldots j_m} X^{j_{m+1}}_{j_1} X^{j_{m+1}}_{j_2}, \ldots, X^{j_{m+1}}_{j_m} \to I_{j_1 j_{m+1}} I_{j_1 j_2 j_{m+1}}, \ldots, I_{j_1 j_2 \ldots j_{m+1}}\}$$

for $1 \leq i \leq N$ and $1 \leq j_1 \leq j_2, \ldots, j_{m+1} \leq t$

In $\sigma_1 = (w_{1,0},\ R_1)$, $w_{1,0} = \lambda$ and
$R_1$:

$$r_{11} = \{T^i_j \to T^i_j,\ go\}$$

$$r_{12} = \{\theta^k \to \beta^k_1 \beta^k_2, \ldots, \beta^k_t\}$$

$$r_{13} = \{T^i_j \beta_j \to \lambda\}$$

$$r_{14} = \{\delta_j \to (I_j)_{go}\}$$

for $1 \leq i \leq N$ and $1 \leq j \leq t$

    In $\sigma_2 = (w_{2,0}, R_2)$, $w_{2,0} = \lambda$ and

    $R_2$:

$$r_{21} = \{I_j T^i_j \rightarrow (I_j T^i_j)_{j'}\}$$

$$r_{22} = \{T^i_j \neg_{I_j} \rightarrow \lambda\}$$

$$r_{23} = \{\xi T^i_{j_1} T^i_{j_2} |\neg_{X^{j_1}_{j_2} X^{j_2}_{j_1}} \rightarrow T^i_{j_1 j_2} |\zeta\}$$

$$r_{24} = \{T^i_{j_1 j_2} \rightarrow T^i_{j_1 j_2}, \, go\}$$

$$r_{25} = \{\theta^k T^i_{j_1 j_2} \rightarrow \beta^k_{j_1 j_2}\}$$

$$r_{26} = \{T^i_{j_1 j_2} \beta_{j_1 j_2} \rightarrow \lambda\}$$

$$r_{27} = \{\delta_{j_1 j_2} \rightarrow (I_{j_1 j_2} X^{j_1}_{j_2} X^{j_2}_{j_1} \xi)_{go}\}$$

for $1 \leq i \leq N$, $1 \leq j_1 \leq j_2 \leq t$ and $1 \leq j' \leq J$

$$\vdots$$

    In $\sigma_h = (w_{h,0}, R_h)$, $w_{h,0} = \lambda$ and

    $R_h$:

$$r_{h1} = \{T_{j_1 j_2 \ldots j_{h-1}} |\neg_{I_{j_1 j_2 \ldots j_{h-1}}} \rightarrow \lambda\}$$

$$r_{h2} = \{\xi T^i_{j_1 j_2 \ldots j_{h-2} j_{h1}} T^i_{j_1 j_2 \ldots j_{h-2} j_{h2}} |\neg_{X^{j_{h1}}_{j_{h2}} X^{j_{h2}}_{j_{h1}}} \rightarrow T_{j_1 j_2 \ldots j_{h-2} j_{h1} j_{h2}} |\zeta\}$$

$$r_{h3} = \{T_{j_1 j_2 \ldots j_{h-2} j_{h1} j_{h2}} \rightarrow T_{j_1 j_2 \ldots j_{h-2} j_{h1} j_{h2}}, \, go\}$$

$$r_{h4} = \{\theta^k T^i_{j_1 j_2 \ldots j_{h-2} j_{h1} j_{h2}} \rightarrow \beta^k_{j_1 j_2 \ldots j_{h-2} j_{h1} j_{h2}}\}$$

$$r_{h5} = \{T^i_{j_1 j_2 \ldots j_{h2}} \beta_{j_1 j_2 \ldots j_{h2}} \rightarrow \lambda\}$$

$$r_{h6} = \{\delta_{j_1 j_2 \ldots j_{h2}} \rightarrow (I_{j_1 j_2 \ldots j_{h2}} X^{j_1}_{j_2} X^{j_2}_{j_1} \xi)_{go}\}$$

for $1 \leq i \leq N$ and $1 \leq j_1 \leq \ldots \leq j_{h2} \leq t$

$$\vdots$$

    In $\sigma_t = (w_{t,0}, R_t)$, $w_{t,0} = \lambda$ and

    $R_t$:

$$r_{t1} = \{T_{j_1 j_2 \ldots j_{t-1}} |\neg_{I_{j_1 j_2 \ldots j_{t-1}}} \rightarrow \lambda\}$$

$$r_{t2} = \{\xi T^i_{j_1 j_2 \ldots j_{t-2} j_{t1}} T^i_{j_1 j_2 \ldots j_{t-2} j_{t2}} |\neg_{X^{j_{t1}}_{j_{t2}} X^{j_{t2}}_{j_{t1}}} \rightarrow T_{j_1 j_2 \ldots j_{t-2} j_{t1} j_{t2}} |\zeta\}$$

$$r_{t3} = \{T_{j_1 j_2 \ldots j_{t-2} j_{t1} j_{t2}} \rightarrow T_{j_1 j_2 \ldots j_{t-2} j_{t1} j_{t2}}, \, go\}$$

$$r_{t4} = \{\theta^k T^i_{j_1 j_2 \ldots j_{t-2} j_{t1} j_{t2}} \rightarrow \beta^k_{j_1 j_2 \ldots j_{t-2} j_{t1} j_{t2}}\}$$

$$r_{t5} = \{T^i_{j_1 j_2 \ldots j_{t2}} \beta_{j_1 j_2 \ldots j_{t2}} \rightarrow \lambda\}$$

$$r_{t6} = \{\delta_{j_1 j_2 \ldots j_{t2}} \rightarrow (I_{j_1 j_2 \ldots j_{t2}} X^{j_1}_{j_2} X^{j_2}_{j_1} \xi)_{go}\}$$

for $1 \leq i \leq N$ and $1 \leq j_1 \leq \ldots \leq j_{t2} \leq t$

    In $\sigma_{t+1} = (w_{t+1,0}, R_{t+1})$, $w_{t+1,0} = \lambda$ and $R_{t+1} = \varnothing$.

    In $\sigma_{j'} = (w_{j,0}, j'_1)$, $w_{j',0} = \lambda$ and

    $R_{j'}$:

$$r_{j'1} = \{T^i_{j'} \rightarrow (T^i_{j'})^{J-1}\}$$

$$r_{j'2} = \{T^i_{j'}, T^i_j \rightarrow \lambda\} \cup \{T^i_{j'}, T^i_j \rightarrow X^j_{j'}\} \cup \{T^i_{j'}, T^i_j \rightarrow X^{j'}_j\}$$

$$r_{j'3} = \{X^{j_1}_{j_2} \rightarrow (\xi X^{j_1}_{j_2})_{go}\}$$

for $1 \le j' \le J$, $1 \le i \le N$ and $1 \le j \le t$.

When computation starts, frequent 1-patterns are generated in cell 1, and then sent to cell 2 and cell 0 by executing rules in parallel. At the same time, objects $T^i_j$ of frequent 1-patterns are also sent to cell 2. Frequent 2-patterns are then generated in cell 2 and sent to cell 3 and cell 0 by executing rules also in parallel, and objects $T^i_j$ of frequent 2-patterns are sent to cell 3. This process stops when all frequent patterns are found. In cell 0, patterns that have subsumes combine with their subsumes to obtain all frequent patterns. When computation ends, the final results are stored in cell 0. Compared to other frequent pattern mining algorithms, the ETPAM algorithm executes evolution rules in parallel to generate frequent patterns, the time complexity of ETPAM is reduced from O($t^2$) to O($t$) as compared with the original Eclat algorithm.

### 3.3. Computing Process

Generation of Frequent 1-Pattern Itemsets. When computation begins, objects $T^i_j$ are entered into cell 1 and then copies of the objects are sent to cell 2 by the rule $r_{11}$. The searching process of the candidate frequent 1-pattern $I_1$ is taken as an example, and the searching processes of the other candidate frequent 1-patterns are similar to that of the process of $I_1$. A total of $k$ copies of $\beta_1$ is generated by rule $r_{12}$ and one $T^i_1$ consumes one $\beta_1$ through rule $r_{13}$. Finally if any copy of $\beta_1$ is left in cell 1, $I_1$ is not a frequent 1-pattern because its support count is less than the threshold $k$; otherwise, if no copy of $\beta_1$ is left in cell 1, $I_1$ is a frequent 1-pattern and is sent to cell 2 and cell 0 through the rule $r_{14}$.

Generation of Subsume of Frequent 1-Patterns. In cell 2, extra objects $T^i_j$ are removed first by rule $r_{21}$, and frequent 1-pattern $I_j$ acts as an inhibitor so that only objects $T^i_j$ belonging to frequent 1-patterns are left in cell 2. Assume that $J$ frequent 1-patterns are obtained in cell 1, then $J$ cells that are the same as cell 2 are generated by rule $r_{22}$. Rule $r_{23}$ cannot be executed without promoter $\xi$. Rules in cell $j'$ for $1 \le j' \le J$ are executed and subsumes of $I_j$ for $1 \le j \le t$ are generated in the corresponding cell $j'$ for $1 \le j' \le J$. In cell $j'$, objects $T^i_{j'}$ belonging to $I_{j'}$ are compared with objects $T^i_j$ belonging to $I_j$ for $1 \le j \le t$ sequentially, and one $T^i_{j'}$ consumes one $T^i_j$. Finally, if both $T^i_{j'}$ and $T^i_j$ remain in the cell, $I_j$ and $I_{j'}$ are not subsume of each other. If just $T^i_j$ remains, $I_j$ is a subsume of $I_{j'}$ and $X^j_{j'}$ is generated by rule $r_{j'2}$. If just $T^i_{j'}$ remains, $I_{j'}$ is a subsume of $I_j$ and $X^{j'}_j$ is generated by rule $r_{j'2}$. In the searching process for subsumes of $I_2$, for example, after $T^3_2 T^6_2 T^8_2 T^9_2$ are compared with $T^3_3 T^6_3 T^8_3 T^9_3$, $T^1_2 T^2_2 T^4_2$, and $T^5_3 T^7_3$ remain, so that $I_2$ and $I_3$ are not subsume of each other. After $T^1_2 T^2_2 T^4_2$ are compared with $T^1_4 T^2_4 T^4_4$, $T^3_2 T^6_2 T^8_2 T^9_2$ remain, so that $I_2$ is a subsume of $I_4$, and $X^2_4$ is generated. When all subsumes of $I_{j'}$ in cell $j'$ are found, computation halts, promoter $\xi$ is generated, and promoter $\xi$ and the subsumes are sent to cell 0 and cell 2.

Generation of Frequent 2-Pattern Itemsets. The execution condition of rule $r_{23}$ is met when promotor $\xi$ appears in cell 2, so that rule $r_{23}$ generates objects $T^i_{j_1 j_2}$ as frequent 2-pattern candidates. Subsumes of frequent 1-patterns are inhibitors of this process, so that only objects $T^i_{j_1 j_2}$ which are not subsumes of each other are generated in cell 2. Duplicates of $T^i_{j_1 j_2}$ are removed by rule $r_{24}$ to keep the uniqueness of the object. Then $T^i_{j_1 j_2}$ left in cell 2 are used to find frequent 2-patterns and one copy of $T^i_{j_1 j_2}$ is sent to cell 3 by rule $r_{25}$. The searching process of candidate frequent 2-pattern $\{I_1 I_2\}$ is taken as an example, and the searching process of other candidate frequent 2-patterns is similar. Totally $k$ copies of $\beta_{12}$ are generated by rule $r_{26}$ and one $T^i_{12}$ consumes one $\beta_{12}$. Finally, if any $\beta_{12}$ remains, $\{I_1 I_2\}$ is not a frequent 2-pattern; otherwise, if no $\beta_{12}$ remains, $\{I_1 I_2\}$ is a frequent 2-pattern and is sent to cell 0 and cell 3 together with subsumes and promotor $\xi$.

Each of the other cells $j$ for $3 \leq j \leq t$ executes evolution rules similar to those in cell 2 and performs similar functions to find $j$-frequent patterns.

In cell 0, $R_0$ is executed to combine frequent patterns obtained with their subsumes to get all frequent patterns of the database. After the algorithm finishes, all frequent patterns are stored in cell 0 as the final results.

### 3.4. Algorithm Specification

The typical Eclat algorithm runs sequentially. However, ETPAM is executed in parallel utilizing the nature of the tissue-like P system. A pseudo code of ETPAM is presented in Algorithm 1 in the following.

---

**Algorithm 1.** ETPAM.

---

**Input:** Transactional database; $\theta^k$ representing the threshold k;
**Method:**
{
Rule $r_{11}$: Transfer one copy of objects $T_j^i$ to cell 2.
Rule $r_{12}$: Generate $\beta_j^k$ for $1 \leq j \leq t$ to check the candidate frequent 1-patterns $T_1$.
Rule $r_{13}$: Check all objects $T_j^i$ in the cell, and one object $T_j^i$ consumes one $\beta_j$. Continue until all objects $T_j^i$ have been checked or all k copies of $\beta_j$ have been consumed.
Rule $r_{14}$: If all k copies of $\beta_j$ have been consumed, generate an object $\delta_j$ to add $I_j$ to $L_1$ as a frequent 1-pattern and transfer $I_j$ cell 2, and cell 0.
Rule $r_{21}$: Generate a new membrane $j'$ for each frequent 1-pattern $I_j$, and transfer the corresponding objects $I_j$ and $T_j^i$ to cell $j'$.
Rule $r_{j'1}$: In cell $j'$, compare objects $T_{j'}^i$ belonging to $I_{j'}$ and objects $T_j^i$ belonging to $I_j$ for $1 \leq j \leq j' \leq t$ in parallel, and one $T_{j'}^i$ consumes one $T_j^i$. Finally, if both $T_{j'}^i$ and $T_j^i$ remain in the cell, $I_j$ and $I_{j'}$ are not subsume of each other. If just $T_j^i$ remains, $I_j$ is a subsume of $I_{j'}$ and then $X_{j'}^j$ is generated. If just $T_{j'}^i$ remains, $I_{j'}$ is subsume of $I_j$ and then $X_j^{j'}$ is generated. Continue this way until all subsumes of $I_{j'}$ have been found.
Rule $r_{j'2}$: Generate object $\xi$ and transfer it to cell 2 and cell 0 together with all subsumes of $I_j$.
For $(2 \leq h \leq t$ and $L_{h-1} \neq \varnothing)$
{
Rule $r_{h1}$: Delete objects $T_{j_1 j_2 \ldots j_{h-1}}^i$ not belonging to frequent $(h-1)$-patterns.
Rule $r_{h2}$: Scan all objects $T_{j_1 j_2 \ldots j_{h-1}}^i$ representing the frequent $(h-1)$-patterns to generate the objects $T_{j_1 j_2 \ldots j_h}^i$ representing the candidate frequent h-patterns $T_h$.
Rule $r_{h3}$: Transfer one copy of objects $T_{j_1 j_2 \ldots j_h}^i$ to cell $h+1$.
Rule $r_{h4}$: Generate $\beta_{j_1 j_2 \ldots j_h}^k$ for each $T_{j_1 j_2 \ldots j_h}^i$ to check the candidate frequent h-patterns $T_h$.
}
Rule $R_0$: combine frequent patterns in cell 0 with their subsumes to get all frequent patterns of database.
}
**Output:** Frequent patterns mined from the database.

---

### 3.5. Time Complexity

In this section, the time complexity of ETPAM in the worst case is evaluated. Obtaining frequent 1-patterns needs 4 steps. Passing objects $T_j^i$ to cell 2 and cell 0 needs 1 step and generating $\beta_j^k$ needs 1 step. Checking all frequent 1-pattern candidates in parallel takes 1 step. Finally sending the frequent 1-patterns obtained in cell 1 to cell 2 and cell 0 needs 1 step.

Obtaining subsumes of frequent 1-patterns needs 5 steps. Generating new membrane $j'$ for each frequent 1-pattern $I_j$ needs 1 step, and transferring the corresponding objects $I_j$ and $T_j^i$ to cell $j'$ needs 1 step. Comparing all $T_{j'}^i$ belonging to $I_{j'}$ and $T_j^i$ belonging to $I_j$ for $1 \leq j \leq j' \leq t$ in parallel and generating subsumes of frequent 1-pattern $I_{j'}$ in cell $j'$ in parallel take 1 step. Executing rules $r_{j'1}$, $r_{j'2}$,

and $r_{j'3}$ in parallel in each cell $j'$ and obtaining subsumes for all frequent 1-patterns simultaneously take 1 step. Sending object $\xi$ and subsumes obtained in cell 1 to cell 2 and cell 0 needs 1 step.

Obtaining frequent h-patterns needs 6 steps. Deleting extra objects $T^i_{j_1 j_2 \ldots j_{h-1}}$ not belonging to frequent $(h-1)$-patterns needs 1 step. Generating candidate frequent h-patterns $T_h$ needs 1 step. Passing objects $T^i_{j_1 j_2 \ldots j_h}$ to cell $h+1$ needs 1 step and generating $\beta^k_{j_1 j_2 \ldots j_h}$ needs 1 step. Checking all frequent h-pattern candidates in parallel takes one step. Finally sending the frequent $h$-patterns obtained in cell h to cell $h+1$ and cell 0 needs 1 step.

Finally, in cell 0, obtaining all frequent patterns by combining all frequent patterns with their subsumes in parallel takes 1 step.

Thus, the complexity of ETPAM is $4 + 4 + 6(t-1) + 1 = 6t + 3$, which gives $O(t)$. Table 2 presents the time complexities of some basic frequent pattern mining algorithms. In the table, $|C_h|$ represents the number of candidate frequent $h$-patterns and $|L_h|$ represents the number of frequent $(h-1)$-patterns. As shown in Table 2, the performance of ETPAM is better than that of other existing algorithms.

**Table 2.** Time complexities of some pattern mining algorithms.

| Algorithm | Time Complexity |
|---|---|
| Apriori [7] | $O(N_t|C_h| + t|L_h - 1|\,|L_h - 1|)$ |
| Parallel Apriori algorithm on Hadoop Cluster [13] | $O(N_t|C_h| + t|L_h - 1|\,|L_h - 1|)$ |
| FP-growth [24] | $O(N_t)$ |
| Eclat [25] | $O(t(t+1))$ |
| dEclat [15] | $O(t(t+1))$ |
| ETPAM | $O(t)$ |

## 4. An Illustrative Example

To give a clear demonstration about how ETPAM works, this section presents an illustrative example to demonstrate the execution of the algorithm using the database in Table 1. As shown in Table 1, the database contains 9 transactions. The threshold is set to $k = 3$.

Generation of Frequent 1-Pattern Itemsets. When computation begins, objects $\{T^1_1 T^4_1 T^5_1\ T^7_1 T^8_1 T^9_1\}$, $\{T^1_2 T^2_2 T^3_2 T^4_2 T^6_2 T^8_2 T^9_2\}$, $\{T^3_3 T^5_3 T^6_3 T^7_3 T^8_3 T^9_3\}$ $\{T^1_4 T^2_4 T^4_4\}$, and $\{T^5_5 T^8_5\}$ are entered into cell 1 and then one copy is sent to cell 2 by rule $r_{11}$. The auxiliary objects $\beta_j$ for $1 \le j \le 5$ are created by rule $r_{12}$. The searching process of candidate frequent 1-pattern $I_1$ is used as an example. Objects $\{T^1_1 T^4_1 T^5_1\}$ and $\{T^7_1 T^8_1 T^9_1\}$ are in cell 1 meaning that item $I_1$ is included in the first, fourth, fifth, seventh, eighth, and ninth transactions. After rules $\{T^1_1 \beta_1 \to \lambda\}$ and $\{T^1_4 \beta_1 \to \lambda\}$ are executed, objects $\{T^5_1 T^7_1 T^8_1 T^9_1\}$ remain in cell 1. Hence, $I_1$ is a frequent 1-pattern, and subrule $\{\delta_1 \to (I_1)_{go}\}$ sends $I_1$ to cell 2 and cell 0. The searching processes of $I_2 I_3 I_4 I_5$ are the same as that of $I_1$. Finally, $\{I_1 I_2 I_3 I_4\}$ are determined to be frequent 1-patterns in cell 1, and are sent to cell 2 and cell 0.

Generation of Subsumes of Frequent 1-Patterns. In cell 2, extra objects $T^1_5 T^8_5$ are removed by rule $r_{21}$, and objects $\{T^1_1 T^4_1 T^5_1 T^7_1 T^8_1 T^9_1\}$, $\{T^1_2 T^2_2 T^3_2 T^4_2 T^6_2 T^8_2 T^9_2\}$, $\{T^3_3 T^5_3 T^6_3 T^7_3 T^8_3 T^9_3\}$, and $\{T^1_4 T^2_4 T^4_4\}$ stay because frequent 1-patterns $I_1 I_2 I_3 I_4$ are inhibitors in rule $r_{21}$. Rule $r_{22}$ is executed to create 4 cells to generate subsumes of frequent 1-patterns. In cell 2′, after $T^1_2 T^4_2 T^8_2 T^9_2$ are compared with $\{T^1_1 T^4_1 T^8_1 T^9_1\}$, $\{T^5_1 T^7_1 T^2_1 T^3_1\}$ and $T^6_2$ remain, so that $I_1$ and $I_2$ are not subsume of each other. After $T^3_2 T^6_2 T^8_2 T^9_2$ are compared with $\{T^3_3 T^6_3 T^8_3 T^9_3\}$, $\{T^1_2 T^2_2 T^4_2 T^5_2\}$ and $T^7_3$ remain, so that $I_2$ and $I_3$ are not subsume of each other either. After $\{T^1_2 T^2_2 T^4_2\}$ are compared with $\{T^1_4 T^2_4 T^4_4\}$, $\{T^3_2 T^6_2 T^8_2 T^9_2\}$ remain, so that $I_2$ is a subsume of $I_4$, and, therefore, $X^2_4$ is generated. To improve efficiency, just subsume indices for items with support larger than the threshold are found. Because $I_5$ is not a frequent 1-pattern, the process of searching subsumes of $I_2$ ends. Object $\xi$, together with $X^2_4$, is generated and sent to cell 2 and cell 0. The searching processes in cells $j'$ for $1 \le j' \le 4$ are similar to that in cell 2′. All rules in cells $j'$ for $1 \le j' \le J$ are executed in parallel, and all frequent 1-patterns' subsumes are obtained simultaneously. The process and results of the generation of frequent 1-patterns are summarized in Table 3.

**Table 3.** Generation of frequent 1-patterns.

| $r_{ij}$ | Cell 0 | Cell 1 |
|---|---|---|
| 0 | | $\{T_1^1 T_1^4 T_1^5 T_1^7 T_1^8 T_1^9\}, \{T_2^1 T_2^2 T_2^3 T_2^4 T_2^6 T_2^8 T_2^9\},$ $\{T_3^3 T_3^5 T_3^6 T_3^7 T_3^8 T_3^9\}, \{T_4^1 T_4^2 T_4^4\} \{T_5^1 T_5^8\}$ |
| 1 | | $\{T_1^1 T_1^4 T_1^5 T_1^7 T_1^8 T_1^9\}, \{T_2^1 T_2^2 T_2^3 T_2^4 T_2^6 T_2^8 T_2^9\},$ $\{T_3^3 T_3^5 T_3^6 T_3^7 T_3^8 T_3^9\}, \{T_4^1 T_4^2 T_4^4\},$ $\{T_5^1 T_5^8\}, \beta_1^3 \beta_2^3 \beta_3^3 \beta_4^3 \beta_5^3, (r_{12})$ |
| 2 | | $\{T_1^7 T_1^8 T_1^9\} \{T_2^4 T_2^6 T_2^8 T_2^9\} \{T_3^7 T_3^8 T_3^9\} \beta_5 (r_{13})$ |
| 3 | $I_1 I_2 I_3 I_4$ | $\{T_1^7 T_1^8 T_1^9\} \{T_2^4 T_2^6 T_2^8 T_2^9\} \{T_3^7 T_3^8 T_3^9\} (r_{14})$ |

Generation of Frequent 2-Pattern Itemsets. Because the execution condition of rule $r_{23}$ is met when promotor $\xi$ appears in cell 2, rule $r_{23}$ generates $\{T_{12}^1 T_{14}^1 T_{12}^4 T_{14}^4 T_{13}^5 T_{13}^7 T_{12}^8 T_{13}^8 T_{12}^9 T_{13}^9 T_{23}^3 T_{23}^6 T_{23}^8 T_{23}^9\}$. Because $X_4^2$ is an inhibitor, no objects like $T_{24}^i$ for items $I_2$ and $I_4$ are generated. Duplicate $T_{j_1 j_2}^i$ are removed by rule $r_{24}$, $T_{j_1 j_2}^i$ left in cell 2 are used to find frequent 2-patterns, and one copy of $T_{j_1 j_2}^i$ is sent to cell 3 by rule $r_{25}$. The auxiliary objects $\beta_{12}^3 \beta_{14}^3 \beta_{13}^3$ and $\beta_{23}^3$ are created by rule $r_{26}$. After $\{T_{12}^1 T_{12}^4\}$ and $T_{12}^8$ are compared with $\beta_{12}$, $T_{12}^9$ remains. After $T_{13}^5 T_{13}^7$ and $T_{13}^8$ are compared with $\beta_{13}$, $T_{13}^9$ remains. After $T_{14}^1$ and $T_{14}^4$ are compared with $\beta_{14}$, $\beta_{14}$ remains. Therefore, $I_{12}$ and $I_{13}$ are frequent 2-patterns. Subrules $\{\delta_{12} \rightarrow (I_{12})_{go}\}$ and $\{\delta_{13} \rightarrow (I_{13})_{go}\}$ send $I_{12} I_{13}$ together with $X_4^2$ and the promotor $\xi$ to cell 0 and cell 3. The process and results of the generation of frequent 2-pattern itemsets are summarized in Table 4.

**Table 4.** Generation of frequent 2-patterns.

| $r_{ij}$ | Cell 0 | Cell 2 |
|---|---|---|
| 3 | $I_1 I_2 I_3 I_4$ | $\{T_1^1 T_1^4 T_1^5 T_1^7 T_1^8 T_1^9\} \{T_2^1 T_2^2 T_2^3 T_2^4 T_2^6 T_2^8 T_2^9\} \{T_3^3 T_3^5 T_3^6 T_3^7 T_3^8 T_3^9\}\{T_4^1 T_4^2 T_4^4\}$ |
| 4 | $I_1 I_2 I_3 I_4 X_4^2$ | $T_{12}^1 T_{14}^1 T_{12}^4 T_{14}^4 T_{13}^5 T_{13}^7 T_{12}^8 T_{13}^8 T_{12}^9 T_{13}^9 T_{23}^3 T_{23}^6 T_{23}^8 T_{23}^9 X_4^2 (r_{23} r_{24})$ |
| 5 | $I_1 I_2 I_3 I_4 X_4^2$ | $T_{12}^1 T_{14}^1 T_{12}^4 T_{14}^4 T_{13}^5 T_{13}^7 T_{12}^8 T_{13}^8 T_{12}^9 T_{13}^9 T_{23}^3 T_{23}^6 T_{23}^8 T_{23}^9$ $\beta_{12}^3 \beta_{14}^3 \beta_{13}^3 \beta_{23}^3 X_4^2 (r_{26})$ |
| 6 | $I_1 I_2 I_3 I_4 X_4^2$ | $T_{12}^9 T_{13}^9 \beta_{14} X_4^2 (r_{27})$ |
| 7 | $I_1 I_2 I_3 I_4 I_{12} I_{13} X_4^2$ | $T_{12}^9 T_{13}^9 X_4^2 (r_{28})$ |

Generation of Frequent 3-Pattern Itemsets. In cell 3, extra objects $\{T_{14}^1 T_{14}^4 T_{23}^3 T_{23}^6 T_{23}^8 T_{23}^9\}$ are removed by rule $r_{31}$ and $\{T_{12}^1 T_{12}^4 T_{13}^5 T_{13}^7 T_{12}^8 T_{13}^8 T_{12}^9 T_{13}^9\}$ stay because frequent 2-patterns $I_{12} I_{13}$ are inhibitors. Because the execution condition of rule $r_{33}$ is met when promotor $\xi$ appears in cell 3, rule $r_{33}$ generates $T_{123}^8$ and $T_{123}^9$. The auxiliary objects $\beta_{123}^3$ are created by rule $r_{36}$. After $T_{123}^8$ and $T_{123}^9$ are compared with $\beta_{123}$, $\beta_{123}$ remains, so that $I_{123}$ is not a frequent 3-pattern. The process ends since there are no frequent 3-patterns. The process and results of the generation of frequent 3-pattern itemsets are summarized in Table 5.

**Table 5.** Exploration process of frequent 3-patterns.

| $r_{ij}$ | Cell 0 | Cell 3 |
|---|---|---|
| 7 | $I_1 I_2 I_3 I_4 \; I_{12} I_{13} \; X_4^2$ | $T_{12}^1 T_{12}^4 T_{13}^5 T_{13}^7 T_{12}^8 T_{13}^8 T_{12}^9 T_{13}^9$ |
| 8 | $I_1 I_2 I_3 I_4 \; I_{12} I_{13} X_4^2$ | $T_{123}^8 T_{123}^9 (r_{32} r_{33})$ |
| 9 | $I_1 I_2 I_3 I_4 \; I_{12} I_{13} X_4^2$ | $T_{123}^8 T_{123}^9 \beta_{123}^3 (r_{34})$ |
| 10 | $I_1 I_2 I_3 I_4 \; I_{12} I_{13} X_4^2$ | $\beta_{123} (r_{35})$ |
| 11 | $I_1 I_2 I_3 I_4 \; I_{12} I_{13} I_{24} I_{124} (r_{36})$ | |

In cell 0, $R_0$ executes to combine $I_2 I_{12}$ with $I_4$ and in turn to generate $I_{24}$ and $I_{124}$. The computation of the P system ends at this point and all frequent patterns $\{I_1 I_2 I_3 I_4 \; I_{12} I_{13} I_{24} I_{124}\}$ are stored in cell 0.

## 5. Experiments

Five databases, Connect, Mushroom, MovieItem, Retail, and T10I4D100K, from the UCI Machine Learning Repository were used in the experiments. Some of these databases are dense and others are sparse, and all of them are often utilized to test the performance of frequent pattern mining methods. The characteristics of these databases are given in Table 6. All experiments were performed on a personal computer with an Intel Core i3 processor and 4 GB of RAM under the Microsoft Windows 10 64-bit operating system. All the programs are coded in Python 3.

**Table 6.** Characteristics of the databases used for the experiments.

| Database | Transactions | Items | Avg. Length |
|---|---|---|---|
| T10I4D100K | 100,000 | 1000 | 10.0 |
| Mushroom | 8124 | 120 | 23.0 |
| Retail | 88,162 | 16,470 | 10.3 |
| MovieItem | 943 | 80,000 | 84.8 |

*5.1. Effectiveness of ETPAM in Identifying the Frequent Pattern Itemsets*

Two databases, Mushroom and Connect, are used to verify the performance of ETPAM in identifying the frequent patterns. The results are reported in the following.

The Mushroom database includes 8124 transactions, each transaction has 23 attributes (fields), each attribute represents one characteristic of the mushrooms, such as the poisonousness of the mushroom, and each attribute has 2 to 12 values for a total of 119 possible values. The purpose is to know what attributes often appear together, i.e., to find frequent patterns in the database. The data is preprocessed first, where each attribute value is treated as a new attribute, and each new attribute has only two values, 1 or 0 representing yes or no. The threshold is set to $k = 4062$ (8124 * 0.50). The frequent patterns found by ETPAM are presented in Table 7.

**Table 7.** Frequent patterns identified by ETPAM for the Mushroom database.

| *h* | Frequent Patterns |
|---|---|
| 1 | {2}{24}{36}{90}{34}{86}{85}{39}{53}{59}{63}{67}{76} |
| 2 | {90, 36}{86, 39}{86, 53}{86, 59}{86, 63}{86, 67}{86, 76}{86, 24}{85, 63}{85, 67}{85, 76}{85, 39}{85, 53} {85, 59}{90, 39}{90, 53}{90, 59}{90, 63}{85, 2}{85, 24}{85, 34}{59, 63}{36, 39}{36, 59}{36, 63}{34, 39} {34, 53}{34, 59}{34, 63}{34, 67}{34, 76}{34, 24}{34, 36}{85, 86}{34, 86}{34, 85}{86, 90}{34, 90}{85, 90} {36, 90}{34, 36}{36, 86}{36, 85}{90, 24} |
| 3 | {34, 36, 85}{90, 36, 39}{90, 36, 59}{86, 90, 24}{86, 90, 39}{86, 90, 53}{86, 90, 59}{86, 90, 63}{85, 86, 90} {34, 85, 90}{36, 85, 86}{85, 36, 39}{85, 36, 59}{85, 36, 63}{86, 34, 39}{86, 34, 53}{86, 34, 59}{86, 34, 63} {86, 34, 67}{86, 34, 76}{86, 34, 24}{85, 86, 24}{85, 59, 63}{85, 34, 24}{34, 90, 24}{34, 90, 36}{34, 90, 39} {34, 90, 53}{34, 90, 59}{34, 90, 63}{34, 36, 39}{34, 36, 59}{34, 85, 86}{34, 86, 90}{34, 36, 90}{85, 90, 39} {85, 90, 53}{85, 90, 59}{85, 90, 63}{85, 90, 24}{85, 86, 39}{85, 86, 53}{85, 86, 59}{85, 86, 63}{85, 86, 67} {85, 86, 76}{85, 86, 36}{36, 86, 90}{36, 85, 90}{85, 34, 39}{85, 34, 53}{85, 34, 59}{85, 34, 63}{85, 34, 67} {85, 34, 76}{34, 36, 86}{86, 36, 39}{86, 36, 59} |
| 4 | {85, 86, 90, 39}{85, 86, 90, 53}{85, 86, 90, 59}{85, 86, 90, 63}{85, 90, 36, 59}{85, 86, 36, 39}{85, 86, 36, 59} {85, 86, 34, 39}{85, 86, 34, 53}{85, 86, 34, 59}{85, 86, 34, 63}{85, 86, 34, 67}{85, 86, 34, 76}{85, 86, 34, 24} {36, 85, 86, 90}{36, 34, 86, 85}{85, 34, 36, 39}{85, 34, 36, 59}{90, 86, 34, 85}{36, 90, 34, 86}{34, 36, 85, 90} {85, 34, 90, 24}{85, 34, 90, 36}{85, 34, 90, 39}{85, 34, 90, 53}{85, 34, 90, 59}{85, 34, 90, 63}{86, 34, 90, 24} {86, 34, 90, 39}{86, 34, 90, 53}{86, 34, 90, 59}{86, 34, 90, 63}{86, 34, 36, 39}{86, 34, 36, 59}{85, 90, 36, 39} {85, 86, 90, 24} |
| 5 | {85, 86, 34, 90, 59}{85, 86, 34, 90, 63}{85, 86, 34, 90, 53}{85, 86, 34, 90, 24} {85, 86, 34, 36, 39}{85, 86, 34, 36, 59}{85, 86, 34, 90, 39} |
| 6 | ∅ |

The Connect database includes 67,557 transactions, each transaction has 43 attributes (fields), and each attribute has 3 values for a total of 129 values. The data is preprocessed in a way similar to that used in the Mushroom database, i.e., each attribute value is treated as a new attribute, and each new attribute has only two values, 1 or 0, representing yes or no. The threshold is set to $k = 66{,}205$ (67,557 * 0.98). The frequent patterns found by ETPAM are presented in Table 8.

**Table 8.** Frequent patterns identified by ETPAM for the Connect database.

| *h* | **Frequent Patterns** |
|---|---|
| 1 | {124}{106}{34}{86}{85}{88}{19}{37}{55}{75}{109}{91}{127} |
| 2 | {109, 91}{37, 91}{109, 55}{55, 75}{106, 37}{106, 55}{106, 75}{106, 127}{106, 91}{106, 109}{19, 88}{37, 88}{55, 88}{75, 88}{19, 37, 75}{127, 19, 37}{124, 75}{109, 124}{124, 91}{124, 127}{106, 19}{127, 88}{109, 88}{88, 91}{19, 37}{19, 55}{19, 75}{127, 19}{109, 19}{19, 91}{37, 55}{37, 75}{127, 37}{109, 37}{127, 55}{55, 91}{127, 75}{109, 75}{75, 91}{109, 127}{127, 91} |
| 3 | {106, 55, 91}{106, 109, 55}{19, 37, 91}{19, 55, 75}{127, 19, 55}{109, 19, 75}{19, 75, 91}{106, 127, 75}{106, 75, 91}{106, 109, 75}{109, 127, 88}{127, 88, 91}{109, 88, 91}{109, 55, 75}{106, 127, 91}{106, 109, 127}{106, 109, 91}{127, 75, 88}{109, 75, 88}{75, 88, 91}{109, 19, 37}{55, 75, 91}{109, 127, 55}{127, 19, 88}{109, 19, 88}{19, 88, 91}{127, 37, 88}{109, 37, 88}{37, 88, 91}{127, 55, 88}{109, 55, 88}{55, 88, 91}{109, 124, 75}{124, 75, 91}{124, 127, 75}{109, 19, 91}{37, 55, 75}{109, 37, 91}{127, 37, 55}{109, 37, 55}{37, 55, 91}{127, 37, 91}{127, 55, 75}{127, 37, 75}{37, 75, 91}{109, 127, 37}{127, 55, 91}{109, 55, 91}{127, 75, 91}{109, 127, 91}{109, 127, 75}{109, 75, 91}{109, 124, 91}{109, 124, 127}{124, 127, 91}{109, 37, 75}{106, 127, 19}{106, 19, 91}{106, 127, 55}{109, 127, 19}{127, 19, 91}{106, 109, 19}{106, 127, 37}{106, 37, 91}{109, 19, 55}{19, 55, 91}{127, 19, 75}{106, 109, 37} |
| 4 | {109, 75, 88, 91}{109, 127, 88, 91}{109, 127, 19, 37}{127, 19, 37, 91}{109, 19, 37, 91}{109, 127, 19, 55}{127, 19, 55, 91}{109, 19, 55, 91}{109, 127, 19, 75}{127, 19, 75, 91}{109, 19, 75, 91}{109, 127, 19, 91}{109, 127, 37, 55}{127, 37, 55, 91}{109, 37, 55, 91}{109, 127, 37, 75}{127, 37, 75, 91}{109, 37, 75, 91}{109, 127, 37, 91}{109, 127, 55, 75}{127, 55, 75, 91}{109, 55, 75, 91}{109, 127, 55, 91}{109, 127, 75, 91}{109, 124, 75, 91}{109, 124, 127, 75}{124, 127, 75, 91}{109, 124, 127, 91}{106, 109, 127, 19}{106, 109, 19, 91}{106, 109, 127, 37}{106, 109, 37, 91}{106, 109, 127, 55}{106, 109, 55, 91}{106, 127, 75, 91}{106, 109, 127, 75}{106, 109, 75, 91}{106, 109, 127, 91}{109, 127, 19, 88}{127, 19, 88, 91}{109, 19, 88, 91}{109, 127, 37, 88}{127, 37, 88, 91}{109, 37, 88, 91}{109, 127, 55, 88}{127, 55, 88, 91}{109, 55, 88, 91}{109, 127, 75, 88}{127, 75, 88, 91} |
| 5 | {109, 127, 37, 55, 91}{109, 127, 37, 75, 91}{109, 127, 19, 37, 91}{109, 127, 55, 75, 91}{109, 124, 127, 75, 91}{106, 109, 127, 75, 91}{109, 127, 19, 88, 91}{109, 127, 37, 88, 91}{109, 127, 55, 88, 91}{109, 127, 75, 88, 91}{109, 127, 19, 55, 91}{109, 127, 19, 75, 91} |
| 6 | ∅ |

## *5.2. Efficiency of the Proposed Algorithm*

To verify the efficiency of the two improvements introduced into the original Eclat algorithm, ETPAM with rules executed serially is used to compare with those of Apriori [7], Fp-growth [24], and the original Eclat algorithm [25]. The total running time is used as a metric to evaluate performance in experiments. The total running time of each algorithm on each database is plotted against the values of the threshold k in Figure 3, where the vertical axis signifies the total running time in seconds and the horizontal axis represents the different threshold values. As shown in Figure 3, ETPAM with rules executed serially is more efficient than Apriori, Fp-growth, and Eclat for all values of the threshold. Thus, the experimental results verify the efficiency of the improvements proposed. More importantly, the evolution rules of the ETPAM algorithm are actually executed in parallel utilizing the nature of tissue-like P system. For example, the process of generating subsumes of the frequent 1-pattern $I_{j'}$ in cell $j'$ is conducted in parallel, rules $r_{j'1}$, $r_{j'2}$, and $r_{j'3}$ are executed in parallel in each cell $j'$, and the subsumes of all frequent 1-patterns are obtained simultaneously. Running in parallel, it will use much less running time, making the algorithm more efficient.

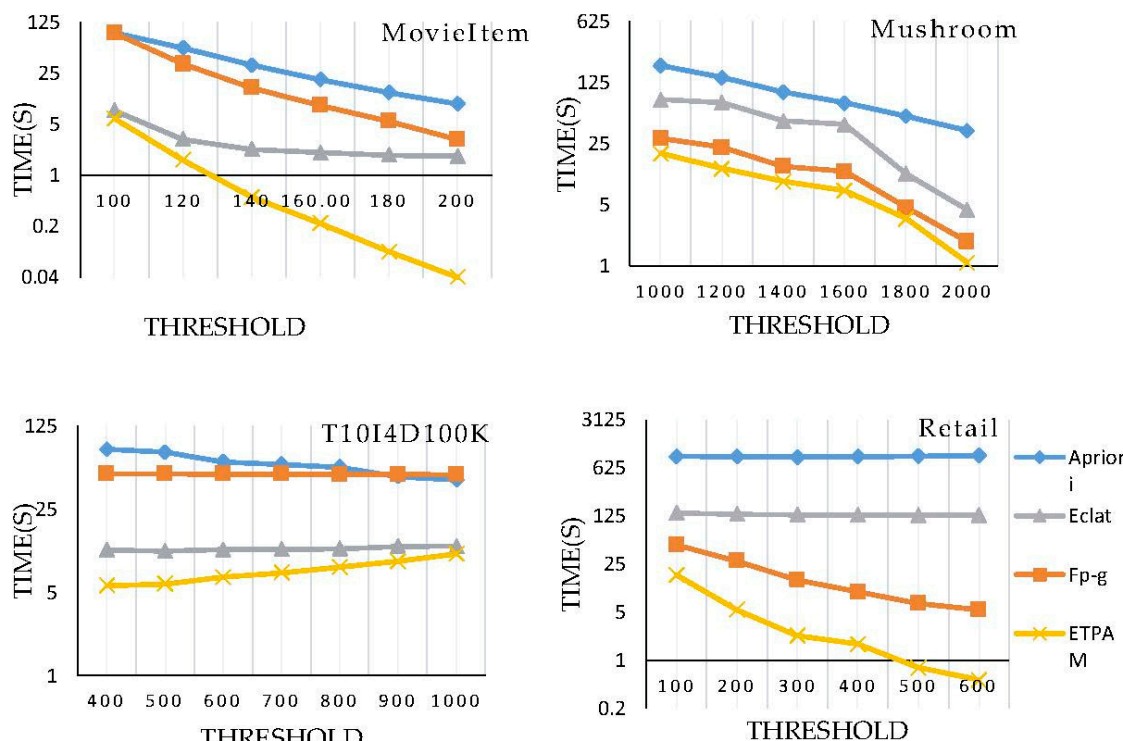

**Figure 3.** Running times of the four algorithms on the four databases.

With these improvements, the time complexity of ETPAM is decreased to O($t$) from O($t^2$) compared to the original Eclat algorithm. The tissue-like P system is a distributed and parallel model, and its evolutionary rules run synchronously, non-deterministically, and in maximum parallel, making the system computationally highly efficient. The tissue-like P system is a natural distributed parallel computing system that can be implemented biologically. The calculation requires only a few cells, which can reduce the computational resource requirements and improve the computational efficiency.

## 6. Conclusions

Membrane computing, inspired by the structure and functioning of biological cells, was introduced as a branch of natural computing. This paper introduces a tissue-like P system with active membranes to mine frequent patterns, and proposes a novel algorithm, called ETPAM, for mining frequent patterns based on the tissue-like P system introduced. ETPAM utilizes the parallel mechanism of the tissue-like P system to execute evolutionary rules synchronously, and in maximum parallel. The time complexity is decreased from O($t^2$) to O($t$) as compared with the original Eclat algorithm. The experimental results using two databases show that ETPAM performed very well in mining frequent patterns. The experimental results on four databases prove that ETPAM is very efficient in mining frequent patterns as compared with three existing algorithms. In addition, only several cells are needed to implement tissue-like P system by biological methods, which can greatly reduce the computing resource consumption. For further research, some other types of P systems, such as the spiking neural P systems (SN P systems) [26] and the cell-like P systems, can be used to develop hopefully more effective and efficient data mining algorithms.

**Author Contributions:** Conceptualization, L.J., L.X. and X.L.; methodology, L.J.; software, L.J.; validation, L.J. and X.L.; formal analysis, L.J.; investigation, L.J.; resources, L.J., L.X. and X.L.; data curation, L.J.; writing—original draft preparation, L.J.; writing—review and editing, L.J. and X.L.; funding acquisition, X.L. and L.X.

**Funding:** This research was funded by the National Natural Science Foundation of China (Nos. 61472231, 61502283, 61876101, 61802234, 61806114).

**Acknowledgments:** This research project is partially supported by the Social Science Foundation of Shandong Province, China (Nos. 16BGLJ06, 11CGLJ22), China Postdoctoral Science Foundation Funded Project (2017M612339, 2018M642695).

**Conflicts of Interest:** The authors declare no conflict of interest.

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
