# Peer review of "An Improved Eclat Algorithm Based on Tissue-Like P System with Active Membranes"

_processes, doi:10.3390/pr7090555_

Round 1
Reviewer 1 Report
Even though the English is very good, some minor changes must be done (listed below). Check the English, check the articles.
Consider swapping and rewriting the paragraphs in the conclusion. First describe the P systems, then the use of the particular system. The very last is the “further research” section.
Correct following errors:
50 are time consuming; -> is time consuming;
52 the computation efficiency -> the computational efficiency
57 generates t + 1 cells -> generates t + 1 cell
64 databases show that -> databases shows that
99 DB and an itemset represents -> DB, and an itemset represents
101 itemset I and TID_set -> itemset I, and TID_set
142 and v means objects -> and \textit{v} means objects
162 of transactions -> of transactions,
182 be transformed to objects -> be transformed into objects
251 all frequent patterns are founded. -> 251 all frequent patterns are found.
258 cell 2 by rule-> cell 2 by the rule
260 copies of \beta_1 are -> copies of \beta_1 is
261 Finally if any copy -> Finally, if any copy
263 otherwise if no -> otherwise, if no
264 rule -> the rule
271 Finally if -> Finally, if
286 Finally if -> Finally, if
299 Finally if both -> Finally, if both
318 Finally in cell 0 -> Finally, in cell 0
391 are dense and other are -> are dense and others are
414 used for the Mushroom -> used in the Mushroom
Reviewer 2 Report
The article concerns a new algorithm called ETPAM, used to search databases and search for patterns found in them. The article presents theoretical foundations and practical application on an example data set. The article is written a bit carelessly. It requires changes in editing and content. I have included in the text of the work bigger editing and factual errors.
Detailed comments:
Errors in markings, symbols and formulas (indicated in the text of the article).
Line 28 "All these P systems have high computing power, are very efficient, and can do what Turing Machines can [4]." - this is a large generalization in the sense of the original definition of the Turing machine, because the amount of data that can be stored and to process computers is over, so for each computer there is only a finite number of programs that it can do.
Line 49-53 - research problems are presented. They are: "(i) The operation of finding the intersection of the TID_sets are time consuming; and (ii) The scale of the TID_sets is quite large and consumes a lot of memory. " The conlucions concern only the first problem. Determining quite a large scale and absorbing large memory is imprecise and has been omitted in the conclusions.
Line 154 - "... introduced by Song et al. [20,23] to restrict the number ... "- both quoted items are not authors Song et al.
The authors use different ways of writing, eg T1,T2,…,Tm or s1,s2,…st+1 or 1,2, ..., m - should be standardized as follows x,. ..., x.
Figure 2 is very simplified and adds nothing to the content of the article. I suggest you remove it or add it in a way that increases its quality.
Line 254 - "... so that the time consumed by the algorithm is not reduced." - The term "substantially reduced" is not appropriate in technical articles. How much? Is this a noticeable shortening? Or maybe it is not worth mentioning because it does not matter for the time of calculations?
Table 2, line 331, Time complexity column - incorrect pattern saving, please correct.
Table 5, last line - error in the record, should be corrected.
Beginning with the Experiments section, tables are incorrectly numbered. It is necessary to correct the numbering of tables and to improve the references to tables in the text of the work.
Line 432 - the application presented in this part does not comply with the provisions in Introduction (line 62) and with the entries in Conclusions (line 445).
The article is very similar with its layout and the approach to the problem posed to the article "An Improved Apriori Algorithm Based on an Evolution-Communication. Tissue-Likeis very similar P System with Promoters and Inhibitors" [7]. This is a bit like a process known as "cutting salami", i.e. publishing articles that arise as a result of sharing the results of scientific research into an unnecessarily large number of scientific publications. This is a discussion remark, but comparing the two articles I have had an impression.

Round 2
Reviewer 2 Report
I recommend the article to print. I suggest you improve Figure 3 (it's illegible - the font size is too small in the legend).
